

# Arsenic transfer along the soil-sclerotium-stroma chain in Chinese cordyceps and the related health risk assessment

YuGuo Liu[*], Ming Shi[*], XiaoShan Liu, JinYing Xie, RunHuang Yang, QiaoWei Ma and LianXian Guo[1]

Dongguan Key Laboratory of Environmental Medicine, School of Public Health, Guangdong Medical University, Dongguan, China

[*] These authors contributed equally to this work.

## ABSTRACT

**Background**. Chinese cordyceps (Lepidoptera: *Ophiocordyceps sinensis*)is a larval-fungus complex. The concentration and distribution of arsenic (As) may vary during the stroma (ST) germination process and between the sclerotium (SC) and the ST. The soil-to-Chinese cordyceps system is an environmental arsenic exposure pathway for humans. We studied the As concentration in the soil, the SC, and the ST of Chinese cordyceps, and performed a risk assessment.

**Methods**. Soil and Chinese cordyceps samples were collected from the Tibetan Plateau in China. The samples were analyzed for the total As concentration and As species determination, which were conducted by inductively coupled plasma mass spectrometry (ICP-MS) and HPLC-ICP-MS, respectively.

**Results**. The concentration of total As in the soil was much higher than in SC and ST. The major As species in the soil was inorganic $As^V$. In SC and ST, organic As was predominant, and the majority of As was an unknown organic form. There are significant differences in the As distribution and composition in soil, SC, and ST. Our risk assessment indicated that chronic daily ingestion was higher than inhalation and dermal exposure in children and adults. The hazard index ($HI$) of the non-carcinogenic and cancer risks ($CR$) for human health were $HI \leq 1$ and $CR < 1 \times 10^{-4}$, respectively.

**Conclusion**. The Chinese cordyceps possesses highly-efficient detoxifying characteristics and has a significant role in As transformation during its life cycle. We found that the levels of As in soils from the habitat of Chinese cordyceps were higher than the soil background values in China, but the probability for incurring health risks remained within the acceptable levels for humans.

Corresponding author
LianXian Guo, glx525@gdmu.edu.cn

## INTRODUCTION

Chinese cordyceps (Fig. 1), a famous fungus, is a fungus-caterpillar complex found mainly in the Tibetan Plateau. The Latin name of this fungus has recently been debated. In this article, we use the phrase ''Chinese cordyceps'' refer to the fungus-caterpillar complex (*Dong et al., 2016*). The Chinese cordyceps goes through two stages to complete its life cycle: teleomorph and anamorph. The ascospores erupt from mature stroma and form directly

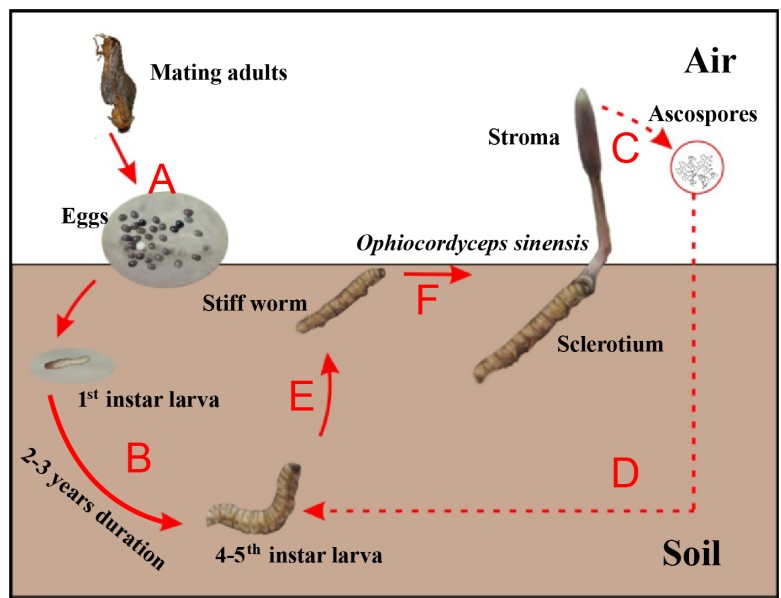

**Figure 1** **Life cycle of Chinese cordyceps.** This figure was modified according to our previous study (*Guo et al., 2018b*). (A) The eggs came out of the insect host and started hatching. (B) The host larvae lived in the soil throughout the long-lasting larval stage. (C) The ascospores were germinated and released from the perithecia. (D) The ascospores infected the 4–5th instar larvae under the ground. (E) The caterpillar filled with threadlike hyphae and formed the sclerotium. (F) The fungus grew out from the head and formed the stroma, Chinese cordyceps finally formed.

into conidia or mycelium in the summer. The conidia or mycelium found in deeper soils in autumn can infect a host (*Zhang et al., 2012*). Similarly, larvae from the host *Thitarodes* (Lepidoptera: *Hepialidae*) infiltrate the soil after incubating from the eggs scattered on the grassland (Fig. 1A) and safely reside in the roots of their preferred substrates throughout the long-lasting larval stage (Fig. 1B) (*Chen et al., 2009*). After developing through four to five instars, which takes approximately two to three years, the larvae may become infected by the fungus in the soil around June (Fig. 1D) (*Zou, Liu & Zhang, 2010*). The larvae then become the fungal host and their interiors are gradually filled with thread-like hyphae, which form the caterpillar-shaped sclerotium (the so-called winter-worm; Fig. 1E) in the winter. The sclerotium germinate from the head of the winter-worm in the spring when the frozen soil thaws and grow into stroma (the so-called summer-grass; Fig. 1F) over approximately 2 months. The stroma mature and disperse millions of spores (Fig. 1C) initiating the next hostile takeover (*Guo et al., 2017*).

Chinese cordyceps has a long history in traditional Chinese medicine. Its pharmaceutical functions are reported to have included antitumor, anti-inflammatory, antioxidant, anti-hyperglycemia, anti-apoptosis, immunoregulatory, and hepatoprotective effects (*Qi et al., 2014*; *Liu et al., 2015*). However, the concentration of As (total As: 4.4–9.0 mg/kg) in Chinese cordyceps was at least three times greater than the reference value of 1 mg/kg (*NHFPC, 2014*), which was disclosed by the China Food and Drug Administration (CFDA) (*CFDA, 2016a*). These levels have raised concerns regarding the health of functional foods

and the promotion of functional foods, specifically Chinese cordyceps, was suspended (*CFDA, 2016b*). The report badly affected the industrial chain of Chinese cordyceps (*Wang, Shan & Sun, 2016*).

Arsenic is an environmental contaminant able to disperse and enter humans through the food chain. It is considered to be the most concerning hazardous material in the world due to its toxicity (*Styblo et al., 2000*). The toxic effect of arsenic depends on its species. Inorganic arsenic (iAs) is carcinogenic to people, as are trivalent iAs (arsenite, $As^{III}$) and pentavalent iAs (arsenate, $iAs^V$), which are widely present in the soil and water (*Huang et al., 2004*). When iAs transfers into organisms along the food chain, it would be transformed into organic arsenic species (oAs) by the organisms. Monomethylarsonic acid ($MMA^V$) and dimethylarsinic acid ($DMA^V$) are the major metabolic products of iAs, which have lower toxicity than iAs. The subsequent metabolites, including organic As compounds: arsenocholine, arsenobetaine (AsB), various arsenolipids, and arsenosugars, are typically considered nontoxic (*Hua et al., 2011*; *Styblo et al., 2000*). Thus, the As transforming processes in organisms are generally detoxifying for iAs. Moreover, some trivalent metabolites, including monomethylarsonous acid ($MMA^{III}$) and dimethylarsenic acid ($DMA^{III}$) in animals and human cells, or arsenic-containing hydrocarbons such as $C_{17}H_{37}AsO$, $C_{19}H_{41}AsO$ and $C_{23}H_{37}AsO$ in seafoods, have been shown to be cytotoxic (*Arroyo-Abad et al., 2010*; *Meyer et al., 2014*). In our previous work (*Guo et al., 2018b*), we found that unknown organic As species (oAsU), which were considered to be arsenosugars, comprised a large proportion of the total As in Chinese cordyceps.

Unlike other mushrooms in which only the stroma or fruiting body is consumed (*Larsen, Hansen & Gössler, 1998*; *Kuehnelt, Goessler & Irgolic, 1997*), the Chinese cordyceps is a larval-fungus complex, and the sclerotium is the complex of the host larva (substrate) and mycelium of the fungus, while the stroma is purely composed of the fungus (*Zhang et al., 2012*). In this context, our study is designed to determine the arsenic species and its distribution in the soil habitat-sclerotium- stroma complex. We studied the risk assessment of As in the soils around the habitat of the Chinese cordyceps on the Qinghai-Tibet Plateau, which has a greater environmental background value of As than other regions in China.

## MATERIALS AND METHODS

### Sample collection and preparation

We selected three sites from the endemic areas in Qinghai-Tibet Plateau for this study. Site A was located at 29°36′N, 94°36′E; Site B was located at 29°35′N, 94°36′E; and Site C, was located at 35°14′N, 91°48′E. We took fifteen soil samples from the 10–20 cm topsoil and twenty Chinese cordyceps samples about 0.3 g each from each sampling site in mid-July 2017. The samples were kept in an icebox and were transported to the laboratory.

In the laboratory, these samples were freeze dried. The Chinese cordyceps samples were divided into two subsamples: SC and ST. ST was light and thin compared with SC, and each of the ten ST subsamples were combined to form a batch sample. Each of the five SC subsamples were combined to form a batch sample. The twenty Chinese cordyceps samples collected from each sampling spot were divided into four batches of sclerotium samples

and two batches of stroma samples, which were named $SC_{A/B/C}$ and $ST_{A/B/C}$ according to sampling sites. Soil samples were ground into powders with a grain size of less than 150 mesh. Every five powdered soil samples were combined into one batch and named *A/B/C* according to sampling sites.

## Sample digestion

0.1 g of each pre-dried sample was digested with concentrated nitric acid (16 mol/L) using the high-temperature and microwave-assisted methods to determine the total As concentration in SC and ST. The digestion methods followed that of our earlier study (*Guo et al., 2018b*). To determine the As speciation, 1 g of the sample powder was digested with 20 mL 0.15 mol/L $HNO_3$ at 90 °C using a water bath for 12 h (*Guo et al., 2018b*). The sample was cooled to room temperature and all of the digested product was centrifuged for 15 min at the speed of 7104 g. The collected supernatant was filtered through a sieve with a mesh aperture of 0.22 $\mu$m and kept in cold storage until analysis.

Approximately 0.1 g of each powdered sample was blended with a mixed solution of hydrochloric acid at 12mol/L and concentrated nitric acid at 16mol/L with the volume ratio of 3:1 to determine the total As concentration in the soil. Digestion was performed according to the standard method, HJ 803-2016 (*MEPP, 2016*). Different As species were extracted according to the method used by *Thomas, Finnie & Williams (1997)*. Briefly, 10 mL 1 mol/L phosphoric acid ($H_3PO_4$) was added into 0.2 g pre-dried sample, processed, and cooled in a microwave. The extract was filtered and diluted with distilled water and prepared for analysis.

## Arsenic determination of sample

The total As was measured by ICP-MS (Agilent 7800, Santa Clara, CA, USA). The separation of As species ($iAs^{III}$, $iAs^V$, MMA, DMA and AsB) were conducted by HPLC (Agilent 1260, Santa Clara, CA, USA) and the separated As species were determined by ICP-MS. Based on our previous study (*Guo et al., 2018b*), the $iAs^{III}$ could not be separated from the other arsenic species. To determine the level of $iAs^{III}$, 1 mL $H_2O_2$ was added into the extraction to fully oxidize the $iAs^{III}$ to $iAs^V$ and the arsenic species were analyzed before (Figs. 2B to 2C, 2F) and after (Figs. 2D to 2E, 2G) $H_2O_2$ was added. The $iAs^{III}$ was calculated by subtracting the level of iAs before addition to $H_2O_2$ from the level of $iAs^V$ after addition to $H_2O_2$.

Each test was performed in triplicate. The concentrations of total As and As species were quantified using calibration curves, which were made with standard samples (National Institute of Metrology, Beijing, China).

The precision of our results was tested by a blank reagent and the Chinese national standard for the green Chinese onion: GBW10049 (GSB-27) and the yellow croaker: GBW08573. Linear responses ranged between 0.5 and 500 $\mu$g/L for the total As determination and between 0.2 and 300 $\mu$g/L for the As species determination; the correlation coefficients were greater than 0.9997 (Table S1). The relative standard deviation (RSD) was less than 10% (Table S2) and the recovery of these certified reference materials was within the acceptable range (Table S3).

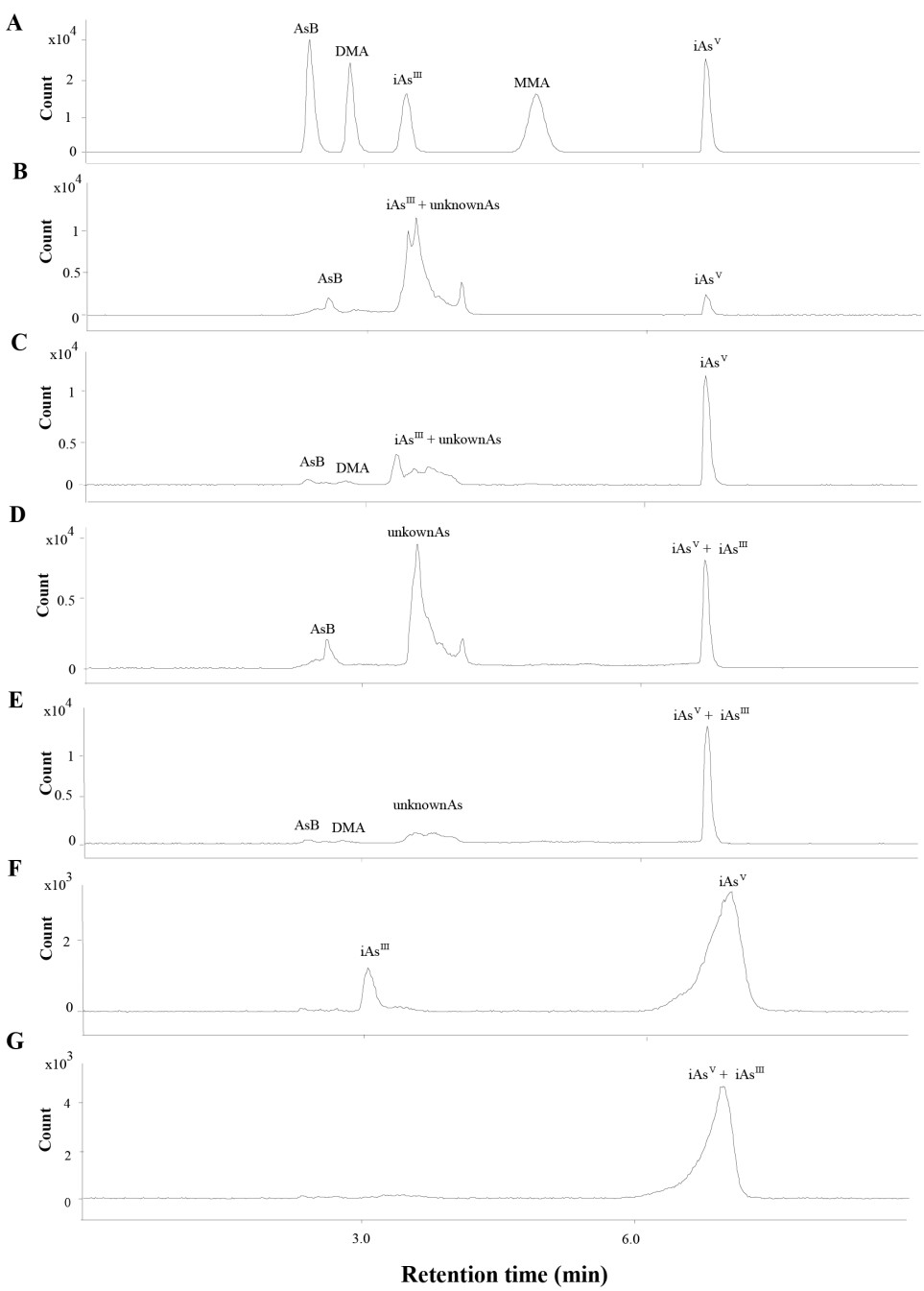

**Figure 2 Chromatograms obtained in quantification by HPLC-ICP-MS.** (A) The mix standards of AsB, DMA, iAs$^{III}$, MMA and iAs$^{V}$, at 50 ppb of each arsenic species. (B & C) Extracts of SC and ST collected from site B, respectively. The iAs$^{III}$ and the other unknown peaks were overlapped. (D & E) Oxidation products of the extracts of SC (B) and ST (C), the iAs$^{III}$ was transformed into iAs$^{V}$ when the extracts was added with $H_2O_2$. (F) The extract of soil sample collected from site B. (G) Oxidation products of the extracts of soil sample (F).

## Health risk assessment of As in soil

Arsenic has been categorized as a chemical carcinogen by *USEPA (2011)*, as well as non-carcinogens for human. Arsenic can migrate into plants and enter the human body through oral ingestion as part of the food chain. The inhalation of soil and dust, and dermal contact are also exposure pathways for humans. Therefore, to comprehensively assess arsenic exposure all three exposure pathways are taken into consideration.

According to the model of human health evaluation by the United States Environment Protection Agency (USEPA) (*USEPA, 1989*), the average daily doses (*ADD*, mg/(kg d)) through the three exposure pathways (ingestion: $ADD_{ing}$; inhalation: $ADD_{inh}$; dermal contact: $ADD_{dermal}$) were calculated separately as follows:

$$ADD_{ing} = C \times \frac{IngR \times EF \times ED}{BW \times AT} \times 10^{-6} \tag{1}$$

$$ADD_{inh} = C \times \frac{InhR \times EF \times ED}{PEF \times BW \times AT} \tag{2}$$

$$ADD_{dermal} = C \times \frac{SA \times AF \times ABF \times EF \times ED}{PEF \times BW \times AT} \times 10^{-6} \tag{3}$$

where $C$ (mg/kg) is the concentration of total As in soil, $EF$ (days/year) is the exposure frequency, $ED$ (years) is the exposure duration, $BW$ (kg) is the body weight, $AT$ (days) is the average time, $PEF$ (m$^3$/kg) is the particular emission factor, $SA$ (cm$^2$) is the surface area of exposed skin, $AF$ (mg/cm$^2$) is the skin adherence factor, $ABF$ is the absorption factor, $IngR$ (mg/d), and $InhR$ (m$^3$/days) is the ingestion rate and inhalation rate, respectively. The parameters for children and adults are shown in Table S4 and refer to the Chinese assessment guidelines for an environmental site (*MEPP, 2014*) and US exposure factors handbook (*USEPA, 2011*).

The hazard quotient (*HQ*) was calculated separately as follows:

$$HQ = \frac{ADD}{RfD} \tag{4}$$

$$HI = \sum HQ_i \tag{5}$$

where $RfD$ is the non-carcinogenic reference dose for As (mg/(kg d)); the values through ingestion, inhalation, and dermal contact are: $3.0 \times 10^{-4}$, $1.5 \times 10^{-5}$, and $3.0 \times 10^{-4}$, respectively (*MEPP, 2014*; *USEPA, 2013*; *Liu et al., 2008*). $HI$ is the total exposure hazard index. If $HQ$ or $HI$ <1, there is no concern for non-carcinogenic effects, whereas potential non-carcinogenic risks may occur in cases where $HQ$ or $HI$ >1.

Carcinogenic risk (*CR*) was calculated as follows:

$$CR = ADD_{ing/inh/dermal} \times SF \tag{6}$$

$$CR_T = \sum CR \tag{7}$$

**Table 1** Concentrations[a] of arsenic speciation (mg/kg) ($n = 15$ for A, B, C, $n = 20$ for SC and ST).

| Sample | Organic arsenic | | | | | Inorganic arsenic | | | tAs |
|---|---|---|---|---|---|---|---|---|---|
| | AsB[b] | DMA | MMA | oAsU | total | As$^{III}$ | As$^V$ | total | |
| A | $0.04 \pm 0.01$ | $0.03 \pm 0.02$ | nd | $3.75 \pm 0.25$ | $3.82 \pm 0.26$ | $0.62 \pm 0.15$ | $8.91 \pm 0.35$ | $9.54 \pm 0.21$ | $16.31 \pm 1.82$ |
| B | $0.03 \pm 0.01$ | $0.04 \pm 0.01$ | nd | $2.06 \pm 0.26$ | $2.13 \pm 0.24$ | $0.74 \pm 0.07$ | $7.46 \pm 0.25$ | $8.20 \pm 0.18$ | $13.03 \pm 1.00$ |
| C | $0.03 \pm 0.01$ | $0.04 \pm 0.03$ | nd | $2.88 \pm 0.29$ | $2.95 \pm 1.01$ | $0.66 \pm 0.03$ | $9.26 \pm 0.39$ | $9.92 \pm 0.37$ | $15.72 \pm 1.57$ |
| SC$_A$ | $0.11 \pm 0.01$ | nd[c] | nd | $5.06 \pm 0.6$ | $5.17 \pm 0.6$ | $0.24 \pm 0.02$ | $0.09 \pm 0.01$ | $0.33 \pm 0.02$ | $5.50 \pm 0.58$ |
| SC$_B$ | $0.10 \pm 0.01$ | nd | nd | $4.22 \pm 0.55$ | $4.32 \pm 0.55$ | $0.23 \pm 0.02$ | $0.09 \pm 0.00$ | $0.32 \pm 0.02$ | $4.64 \pm 0.54$ |
| SC$_C$ | $0.13 \pm 0.02$ | nd | nd | $5.14 \pm 0.42$ | $5.27 \pm 0.43$ | $0.31 \pm 0.01$ | $0.1 \pm 0.01$ | $0.41 \pm 0.01$ | $5.68 \pm 0.44$ |
| ST$_A$ | nd | nd | nd | 0.77 | 0.77 | 0.07 | 0.19 | 0.26 | 1.03 |
| ST$_B$ | 0.01 | 0.01 | nd | 0.85 | 0.87 | 0.07 | 0.19 | 0.26 | 1.13 |
| ST$_C$ | nd | nd | nd | 0.63 | 0.63 | 0.07 | 0.12 | 0.19 | 0.82 |

**Notes.**

[a]Concentrations were presented in SC$_{A/B/C}$ and soil (A/B/C) as mean standard deviation, concentrations were presented in ST$_{A/B/C}$ as the average value.

[b]AsB, DMA, MMA, oAsU, As$^{III}$, As$^V$ and tAs were the abbreviation of arsenobetaine, dimethylarsenic acid, monomethylarsonic acid, unknown organic arsenic, arsenite, arsenate, and total arsenic, respectively.

[c]not detected.

where $SF$ is the slope factor of As and the values through ingestion, inhalation, and dermal contact is: 1.5, $4.3 \times 10^{-3}$, and 1.5, respectively (*USEPA, 2011*; *MEPP, 2014*; *USEPA, 2013*; *Liu et al., 2008*). $CR_T$ is the sum of $CR$ for the three pathways. The probability of cancer risk for humans over a lifetime is characterized by $CR$ with an acceptable range from $1.0 \times 10^{-6}$ to $1.0 \times 10^{-4}$. If $CR < 1.0 \times 10^{-6}$, which suggests no significant effect on human beings; $CR > 1.0 \times 10^{-4}$ is likely to be harmful to humans.

## Statistical analysis

Data were analysed using Microsoft Excel 2013 (Microsoft, Redmond, WA, USA) and SPSS 13.0 (IBM, Chicago, IL, USA). The levels of As were calculated as the means ± standard deviations (SD). Wilcoxon and Kruskal-Wallis tests were used to check the significance in the concentrations of total As and As species among different samples. The significant difference was considered to be $p < 0.05$.

## RESULTS

### Total arsenic concentration

The concentrations of total As in the soil samples are presented in Table 1. The highest level of total As was shown in Site A (16.31 mg/kg) and the lowest level was shown in Site B (13.03 mg/kg). The mean concentrations of total As in Sites A, B and C were 1.5, 1.2, and 1.4 times higher than the background soil values in China, respectively (*Wei et al., 1991*).

The concentrations of total As in SC and ST are reported in Table 1. The mean concentration of total As in SC from the study area was between 4.64 and 5.68 mg/kg. By comparing the reference value of total As in functional foods (*NHFPC, 2014*), it was observed that total As in SC was about five times higher. The mean level of total As in ST ranged from 0.82 to 1.13 mg/kg, which was close to the reference value (*NHFPC, 2014*).

The concentration of total As decreased as follows: soil >SC > ST ($p < 0.01$, Wilcoxon and Kruskal–Wallis tests, Table S5).

**Table 2  Non-carcinogenic average daily exposure doses of As in soil (mg/kg d).**

| Site | $ADD_{ing}$ | | $ADD_{inh}$ | | $ADD_{dermal}$ | | $ADD_{total}$ | |
|------|--------|----------|--------|----------|--------|----------|--------|----------|
|      | Adults | Children | Adults | Children | Adults | Children | Adults | Children |
| A | 2.8 E−05 | 2.0 E−04 | 3.2 E−09 | 5.6 E−09 | 7.0 E−10 | 3.2 E−10 | 2.8 E−05 | 2.0 E−04 |
| B | 2.2 E−05 | 1.6 E−04 | 2.5 E−09 | 4.5 E−09 | 5.6 E−10 | 2.5 E−10 | 2.2 E−05 | 1.6 E−04 |
| C | 2.7 E−05 | 1.9 E−04 | 3.1 E−09 | 5.4 E−09 | 6.7 E−10 | 3.0 E−10 | 2.7 E−05 | 1.9 E−04 |

## Arsenic species

The concentrations of different As species in soil samples are shown in Table 1 (the chromatograms are shown in Figs. 2F to 2G). The results showed that inorganic As was abundant in Sites A, B, and C, and iAs$^{V}$ was significantly higher than iAs$^{III}$ (Table S6). The concentration of organic As was significantly lower than inorganic As (Table S6), and small amounts of AsB were detected in organic As.

The concentrations of different As species in the SC and ST samples were presented in Table 1 (the chromatograms are shown in Fig. 2). Under the $H_2O_2$ treatment, most of the As species in the large peak area were not oxidized to iAs$^{V}$ (Figs. 2D to 2E), which proved that the major overlapped peak was not the toxic iAs$^{III}$ but various unknown organic As species (oAsU). However, it was not possible to evaluate their definite compounds and structures due to the lack of appropriate standards.

Inorganic As was in the minority in the SC samples, in which iAs$^{III}$ was significantly higher than iAs$^{V}$ (Table S6). The concentration of organic As was significantly higher when compared with inorganic As. Among these organic As, oAsU was abundant, while DMA and MMA were almost negligible. In the ST samples, inorganic As was also in the minority, while iAs was significantly higher than iAs$^{III}$ (Table S6). The concentration of organic As was significantly higher than inorganic As (Table S6). Among the detected organic As species, oAsU was the predominant species and AsB and DMA were detected in minor amounts in some samples.

## Hazard assessment of the soil

The calculated average daily doses (ADD) for non-carcinogens and carcinogens are summarized in Tables 2 and 3. The ADD decreased through different exposure pathways in the following order: ADD$_{ing}$ >ADD$_{inh}$ >ADD$_{derm}$, indicating that ingestion is the major exposure pathway. Children are more vulnerable to toxicity than adults because of the higher ADD. The results of human health risk assessment of As in soil suggested that the potential non-carcinogenic risk was negligible since the HI was less than 1 (Table 4). The cancer risks (CR) for human health were at an acceptable level (total CR<$1 \times 10^{-4}$) (Table 4).

# DISCUSSION

## Arsenic transfer chain during Chinese cordyceps formation

As a special organism growing in the Tibetan Plateau (Li et al., 2008), Chinese cordyceps is considered to be a consumer and a de-composer in the food-chain. Our study revealed

**Table 3  Carcinogenic average daily exposure doses of As in soil (mg/kg d).**

| Site | $ADD_{ing}$ | | $ADD_{inh}$ | | $ADD_{dermal}$ | | $ADD_{total}$ | |
|---|---|---|---|---|---|---|---|---|
| | Adults | Children | Adults | Children | Adults | Children | Adults | Children |
| A | 9.6 E−06 | 1.7 E−05 | 1.1 E−09 | 4.8 E−10 | 2.4 E−10 | 2.7 E−11 | 9.6 E−06 | 1.7 E−05 |
| B | 7.7 E−06 | 1.3 E−05 | 1.7 E−09 | 3.8 E−10 | 1.9 E−10 | 2.2 E−11 | 7.7 E−06 | 1.3 E−05 |
| C | 9.2 E−06 | 1.6 E−05 | 1.1 E−09 | 4.6 E−10 | 2.3 E−10 | 2.6 E−11 | 9.2 E−06 | 1.6 E−05 |

**Table 4  Index of carcinogenic risk and non-carcinogenic risk.**

| Site | Groups | $HQ_{ing}$ | $HQ_{inh}$ | $HQ_{dermal}$ | HI | $CR_{ing}$ | $CR_{inh}$ | $CR_{dermal}$ | $CR_T$ |
|---|---|---|---|---|---|---|---|---|---|
| A | Adults | 0.09 | 2.59E−05 | 2.32E−06 | 0.09 | 1.4E−05 | 4.69E−12 | 3.58E−10 | 1.4E−05 |
| | Children | 0.66 | 4.54E−05 | 1.05E−06 | 0.66 | 2.5E−05 | 2.06E−12 | 4.05E−11 | 2.5E−05 |
| B | Adults | 0.07 | 2.07E−05 | 1.85E−06 | 0.07 | 1.1E−05 | 3.74E−12 | 2.86E−10 | 1.1E−05 |
| | Children | 0.52 | 3.63E−05 | 8.38E−07 | 0.52 | 2.0E−05 | 1.65E−12 | 3.23E−11 | 2.0E−05 |
| C | Adults | 0.09 | 2.49E−05 | 2.23E−06 | 0.09 | 1.4E−05 | 4.52E−12 | 3.45E−10 | 1.4E−05 |
| | Children | 0.63 | 4.38E−05 | 1.01E−06 | 0.63 | 2.4E−05 | 1.99E−12 | 3.90E−11 | 2.4E−05 |

the total As abundance in the soil of the habitat of the Chinese cordyceps was higher than the average overall abundance. Furthermore, inorganic As accounted for the majority of total As in soil. The original organisms in this soil ecosphere have developed detoxifying strategies to survive and adapt to the toxic circumstances. Although the As transfer chain and corresponding metabolism from the soil to Chinese cordyceps have not been investigated, previous research on plants (*Zhao et al., 2009*; *Lomax et al., 2012*), animals (*Healy et al., 1998*), and fungi (*Gonzálvez et al., 2009*; *Soeroes et al., 2005*; *Chang et al., 2019*) may explain the complicated delivery and transformation of As as follows: first, through the passive absorption from plants' roots, the original As in the soil is transported and isolated into the plant vacuoles to avoid its toxic effects. During this process, the inorganic As keeps its original speciation because plant cells cannot regulate the methylation of As due to lack of methyltransferase (*Zhao et al., 2009*; *Lomax et al., 2012*). The host *Thitarodes* larvae, which take the plants' tender roots for two to three years as their preferred food (Fig. 1B), first reduce the ingested iAs$^V$ to iAs$^{III}$ by their reductase and then methylate iAs$^{III}$ to low toxic MMA or DMA via methylationase. Subsequently, MMA and DMA are detoxified into other nontoxic As compounds. Notably, fungus also contain methylationase for the methylation (*Tang et al., 2016*; *Zhang et al., 2017*). We found that both the As concentration and speciation were significantly different between the soil environment and SC. The larva-fungi union may have highly efficient detoxifying mechanisms through which the inorganic As ingested by *Thitarodes* larvae had been turned into organic As.

It was not possible to accurately evaluate the effect of Chinese cordyceps on As transformation based on changes in the SC since SC was the complex of host larvae and mycelium. Thus, we focused on the concentration and distribution of As across the ST, which grew only from the Chinese cordyceps without any interference from the host tissue. Here we found that the level of total As from SC to ST has been reduced greatly. The level of iAs$^{III}$ was significantly higher than that of iAs$^V$ in the SC, but it was the opposite in

the ST. The results provided strong evidence that although the host larvae ingested large amounts of toxic $iAs^{III}$ from the soil due to $iAs^{III}$ solubility (*Andrahennadi & Pickering, 2008*), Chinese cordyceps can turn substantial parts into low toxic $iAs^{V}$ to prevent toxicity to offspring (ascospores in ST).

The cultivation of wild Chinese cordyceps, which occurred from *Thitarods* in the habitat's natural soil has not been successful because the occurrence mechanism has been unknown. Artificial laboratory cultivation was based on the cultivated *Thitarods* fed with prepared feed containing a low As background, and its life span of six months was much shorter than wild Chinese cordyceps (two to three years). Our previous study (*Guo et al., 2018a*) compared the total As and As species in wild Chinese cordyceps and cultivated Chinese cordyceps. The cultivated Chinese cordyceps were bred under artificial circumstance with trace As in place of the high concentrations of As, which occur naturally on the Tibetan Plateau. Our results showed that As concentration in the cultivated Chinese cordyceps was much lower than that in wild Chinese cordyceps. This finding provided important evidence that the species and As level were affected by the comprehensive function of soils, host larvae, and Chinese cordyceps fungus for wild Chinese cordyceps. It may be inferred, based on the previous study and the results of this experiment, that unlike *Laccaria amethystea* (*Larsen, Hansen & Gössler, 1998*) and *Collybia butyracea* (*Kuehnelt, Goessler & Irgolic, 1997*) which can accumulate As, Chinese cordyceps can reduce As.

## Arsenic concentration in soil and health risk assessment

We found that the total As concentration in soil samples measured by ICP-MS was much higher than the sum of the five As species measured by HPLC-ICP-MS. The difference between the two was unextracted arsenic ores (*Liu et al., 2018*). Therefore, inorganic arsenic was the predominant form found in the soil and so we took the concentration of total As to assess the potential risk posed by soil arsenic. A previous study reported that the soil's As level in Lhasa was higher than that in our tested sites (*Cheng et al., 2014*) and the elevated As concentration may be related to transportation pollutants in addition to the local background values.

Arsenic can exist in almost all environmental media, especially in the soil. It can accumulate in plants and eventually sneak into the body through the food chain (*Wei et al., 2016*; *Tsuda et al., 1992*). Animal husbandry and the dairy industry have long occupied the important position in the local economy where this study was conducted. Arsenic can pose significant health risks through the soil-plants-food-human pathway. However, there was no serious threat to human health based on our results, although As geological background value was higher than that in China. It is worth noting that children were generally more susceptible than adults, which is consistent with many other studies (*Chen et al., 2019*; *Li et al., 2018*). However, due to the toxicity variations of As species, further studies should focus on the potential risk caused by toxic As species rather than the total As.

## CONCLUSIONS

We found that the distribution and species of As were varied among the habitat soil, SC, and ST, suggesting that Chinese cordyceps was not an As-accumulating fungus, as

traditionally believed. In addition, we explained the process of arsenic degradation and translocation. Overall, this study provides a new insight into the detoxification mechanism of Chinese cordyceps under high As stress and can be beneficial to the revival of the Chinese cordyceps-dependent industry. Our risk assessment found that there was little risk for humans caused by As in the high geological background area of Qinghai-Tibet Plateau. In order to provide more evidence, there should be additional research to determine the potential risk caused by different arsenic species.

### Funding
This work was supported by the National Natural Science Foundation of China (No. 81303155), the Natural Science Foundation of Guangdong Province (No. 2018A030313094, 2020A151501457), the Social Science and Technology Development Project of Dongguan (No. 20185071521641), the Public Health and Preventive Medicine Discipline Development Funds of Guangdong Medical University in 2020 (No. 4SG20003G), the Talents Recruitment Program of Guangdong Medical University (No. 4SG19003Gd), and the Science and the Technology Key Project of Zhangjiang (No. 2017B01233). The funders had no role in study design, data collection and analysis, decision to publish, or preparation of the manuscript.

### Grant Disclosures
The following grant information was disclosed by the authors:
National Natural Science Foundation of China: No. 81303155.
Natural Science Foundation of Guangdong Province: No. 2018A030313094, 2020A151501457.
Social Science and Technology Development Project of Dongguan: No. 20185071521641.
Public Health and Preventive Medicine Discipline Development Funds of Guangdong Medical University in 2020: No. 4SG20003G.
Science and the Technology Key Project of Zhangjiang:  No. 2017B01233.

### Competing Interests
The authors declare there are no competing interests.

### Author Contributions
- YuGuo Liu performed the experiments, prepared figures and/or tables, authored or reviewed drafts of the paper, and approved the final draft.
- Ming Shi conceived and designed the experiments, prepared figures and/or tables, and approved the final draft.
- XiaoShan Liu analyzed the data, authored or reviewed drafts of the paper, and approved the final draft.
- JinYing Xie and RunHuang Yang analyzed the data, prepared figures and/or tables, and approved the final draft.

- QiaoWei Ma performed the experiments, prepared figures and/or tables, and approved the final draft.
- LianXian Guo conceived and designed the experiments, prepared figures and/or tables, authored or reviewed drafts of the paper, and approved the final draft.

## Data Availability

Raw data is available in the Supplemental Files.

## Supplemental Information

Supplemental information for this article can be found online at http://dx.doi.org/10.7717/peerj.11023#supplemental-information.

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
