# Peer review of "Arsenic transfer along the soil-sclerotium-stroma chain in Chinese cordyceps and the related health risk assessment"

_PeerJ, doi:10.7717/peerj.11023_

## Round 0.1 · original submission · Major Revisions

It will be critical for your revised manuscript to include statistical analyses to support the conclusions you have presented. Please pay particular attention to this and other reviewer comments. When you have completed your revisions, please review the manuscript carefully for grammar and/or consider using an editing service to improve the readability.

Reviewer 1 ·

Basic reporting

The authors may improve the clarity and discussion of the paper and send this paper to an English-speaking editor.

Experimental design

Authors did not perform statistical analysis

Validity of the findings

'no comment'

Additional comments

This paper presents information regarding arsenic speciation in tissues of Chinese cordyceps, habit soils, and its risk assessment. Authors found that this fungus was not an arsenic accumulator and the results of risk assessment at acceptable levels.
The topic of the manuscript is interesting. However, in its current state, it is not ready. The authors may improve the clarity and discussion of the paper and send this paper to an English-speaking editor. It is missing statistical analysis and quality control.

Introduction
Line 99: use (Huan et al., 2011) instead of (Huan and Carew, 2011). Line 103: explain and give examples of As hydrocarbons. Line 105: move “in Chinese cordyceps” (Line 107) after "(4.00 to 5.25 mg/kg)". Line 108: include a coma after -MS. Line: 116 use “finally,” instead of "ultimately". Line 117: use “determine” instead of "clarify". Line 120-124: this section may improve.

Materials and methods
Line 128: use “endemic areas” instead of "native habits". Line 130: begin the sentence without "And". Line 130: indicate the soil sample number. Line 132: include “samples” after "cordyceps". Lines 132-138: authors may improve this paragraph. Line 145 it is missing the procedure digestion.
Line 151: eliminate “it was”, and use "at" instead of “to”. Lines 152 – 158: this paragraph may be improved, please check the use of prepositions. Line 156: add the ascorbic acid formula. Line 158: add the equivalence 8000 rpm in (g), and use "filtered" instead of "passed". Line 162: authors may improve this sentence. Lines 171-173: define AsB, MMAV, and DMAV. Line 82: Authors may include information regarding the standards used for quality control, % recovery, coefficient of variation, limit of detection. The samples were analyzed by duplicate? Line 187: use “were” instead of "are". Line 199: Is this the correct citation?.

Results
Lines 224-225: Authors may explain this sentence further. Line 227: "National, 2014" is a correct citation. Line 231: use "(Wei et al., 1991)" instead of "(Wei and Zheng, 1991)". Lines 262-276: Authors may improve these sentences. Line 281: Authors may indicate if HQ and CR were calculated using total or extracted arsenic.

Discussion
Line 302: include proportion. Line 314: eliminate “reduced”. Line 325: was a statistical determination conducted? Line 329 and 333: include “total” before "As". Line 335: include “lower” before "that found" and change "that" for "than those found in..." Lines 336-339: Authors may improve this sentence, also figure 3A is missing the black dashed line. Line 341: figures 2B and 3C are missing a red dashed line. Lines 342-345: authors may improve this sentence. Line 347: include “total” before "As in". Lines 347-371: authors may improve this section.

References
Line 418: Should this reference (CFDA, 2016) be quoted in this manner? Reference (2016b) is also missing. Line 421: Should this reference (CFDA, 2016a) be quoted in this manner?. Line 453-457: authors may check the correct way of citing these references.

Tables
Table 1: Include the number of samples for SC, ST, and A, B, C. Tables 2 and 3 are missing HQ and CR results.

Figures
Fig. 2- it is possible that As was not properly separated?
Figure 3A is missing the black dashed line.
Figures 3B and 3C are missing red dashed line.

·

Basic reporting

Review of the manuscript: PeerJ #51522
Title: Arsenic transfer chain in soil-sclerotium-stroma of Chinese cordyceps and its health risk assessment for human exposure via soil
The authors measured the As concentrations in the soil and in the various part of the endoparasitoid fungus Chinenesis cordyceps to assess the toxicological and carcinogenic risk associated to the consumption of this fungus that is wide spread in the Chinese medicine. They also determine the different species of as in the different part of the fungus using HPLC-ICP-MS. They found that the fungus dilutes and detoxifies the As from sol to sclerotium to stroma. One drawback is the absence of statistics analysis of the data in the tables and figures, the authors should perform the ANOVA with a post hoch test on those numbers to strengthen their results.
The study is very interesting and well explained. Only few questions:
1)from where did the standards of arsenic for the analyses come from?
2) cloud the author clarify the equations 1,2, 3 explaining the meaning of all the parts of the equations?
3) Do the authors have an explanation theory about why the ratio of iAs/oAs and AsV/AsIII increase from sclrerotium to stroma whilst in the stroma there is less of all the As species in comparison to the stroma?
The figures and tables are of good quality and readable. The bibliography is in keeping with the subject of the manuscript.
Review of the manuscript: PeerJ #51522
Title: Arsenic transfer chain in soil-sclerotium-stroma of Chinese cordyceps and its health risk assessment for human exposure via soil
The authors measured the As concentrations in the soil and in the various part of the endoparasitoid fungus Chinenesis cordyceps to assess the toxicological and carcinogenic risk associated to the consumption of this fungus that is wide spread in the Chinese medicine. They also determine the different species of as in the different part of the fungus using HPLC-ICP-MS. They found that the fungus dilutes and detoxifies the As from sol to sclerotium to stroma. One drawback is the absence of statistics analysis of the data in the tables and figures, the authors should perform the ANOVA with a post hoch test on those numbers to strengthen their results.
The study is very interesting and well explained. Only few questions:
1)from where did the standards of arsenic for the analyses come from?
2) cloud the author clarify the equations 1,2, 3 explaining the meaning of all the parts of the equations?
3) Do the authors have an explanation theory about why the ratio of iAs/oAs and AsV/AsIII increase from sclrerotium to stroma whilst in the stroma there is less of all the As species in comparison to the stroma?
The figures and tables are of good quality and readable. The bibliography is in keeping with the subject of the manuscript.

Experimental design

No comment

Validity of the findings

1)from where did the standards of arsenic for the analyses come from?
2) cloud the author clarify the equations 1,2, 3 explaining the meaning of all the parts of the equations?
3) Do the authors have an explanation theory about why the ratio of iAs/oAs and AsV/AsIII increase from sclrerotium to stroma whilst in the stroma there is less of all the As species in comparison to the stroma?
The figures and tables are of good quality and readable. The bibliography is in keeping with the subject of the manuscript.

Additional comments

The authors measured the As concentrations in the soil and in the various part of the endoparasitoid fungus Chinenesis cordyceps to assess the toxicological and carcinogenic risk associated to the consumption of this fungus that is wide spread in the Chinese medicine. They also determine the different species of as in the different part of the fungus using HPLC-ICP-MS. They found that the fungus dilutes and detoxifies the As from sol to sclerotium to stroma. One drawback is the absence of statistics analysis of the data in the tables and figures, the authors should perform the ANOVA with a post hoch test on those numbers to strengthen their results.
The study is very interesting and well explained. Only few questions:
1)from where did the standards of arsenic for the analyses come from?
2) cloud the author clarify the equations 1,2, 3 explaining the meaning of all the parts of the equations?
3) Do the authors have an explanation theory about why the ratio of iAs/oAs and AsV/AsIII increase from sclrerotium to stroma whilst in the stroma there is less of all the As species in comparison to the stroma?
The figures and tables are of good quality and readable. The bibliography is in keeping with the subject of the manuscript.

---

## Round 0.2 · Minor Revisions

Your revised document is much improved. However, there are still issues with grammar as identified by Reviewer 1. Please review the manuscript carefully (again) for grammar and/or consider using an editing service to improve the readability.

Reviewer 1 ·

Basic reporting

The new document is better, and the authors have included the recommendations. However, the quality of English grammar should be improved. Some examples where the language could be improved include lines 37-38, 40-41, 42-44, 88, 90-92, 120, 137-138, 148-149, 155, 162-164, 204-205, and 245.

Experimental design

'no comment'

Validity of the findings

'no comment'

Additional comments

The new document is better, and the authors have included the recommendations. However, the quality of English grammar should be improved. Some examples where the language could be improved include lines 37-38, 40-41, 42-44, 88, 90-92, 120, 137-138, 148-149, 155, 162-164, 204-205, and 245.

---

## Round 0.3 · Minor Revisions

Thank you for your revised manuscript. In reviewing your responses to reviewer comments as well as the revised version, I remain concerned about the grammar. This is a final ultimatum to have the document correctly edited or it will be rejected on the next submission.

---

## Round 0.4 · accepted · Accept

Thank you for improving the readability of your work.